# MetaInv: Overcoming Iterative and Direct Method Limitations for Inverse Learning

## Abstract

Invertible neural networks (INNs) have gained significant traction in tasks requiring reliable bidirectional inferences, such as data encryption, scientific computing, and real-time control. However, iterative methods like i-ResNet face notable limitations, including instability on non-contractive mappings and failure in scenarios requiring strict one-to-one mappings. In contrast, analytical approaches like DipDNN guarantee invertibility but at the expense of performance, particularly in tasks demanding rich feature extraction (e.g., convolutional operations in complex image processing). This work presents a detailed analysis of the limitations in current invertible architectures, examining the trade-offs between iterative and analytical approaches. We identify key failure modes, particularly when handling information redundancy or strict bijections, and propose a meta-inverse framework that dynamically combines the advantages of both i-ResNet and DipDNN. Our framework adapts in real-time based on task-specific signals, ensuring both flexibility and guaranteed invertibility. Extensive experiments across diverse domains demonstrate that our hybrid approach outperforms existing methods in forward accuracy, inverse consistency, and computational efficiency. Our results highlight the utility of this meta-inverse strategy for critical applications where precision, stability, and adaptability are crucial.

## 1 Introduction

In recent years, invertible neural networks (INNs) have become essential for a broad range of applications that require consistent bidirectional inference, such as data encryption, scientific computing, and real-time control systems Raissi et al. (2019); Devlin et al. (2019). In these settings, the ability to precisely compute forward and inverse mappings is crucial for tasks like data recovery, state estimation, and adaptive control. Ensuring robust, accurate inverse computations underlies the reliability of these applications Levine et al. (2016); Lewis et al. (2012).

While iterative approaches such as i-ResNet Behrmann et al. (2019) offer flexibility and have been successful in complex, high-dimensional mappings (e.g., image processing and generative models), they come with significant drawbacks. i-ResNet depends on the Lipschitz condition for convergence, which is often difficult to guarantee in practice, especially in tasks requiring strict one-to-one mappings or non-redundant information. Furthermore, i-ResNet suffers from performance issues, particularly when the mappings are non-contractive, leading to unstable or inaccurate inverse estimations Zhang et al. (2020a). In some cases, it may even produce outputs that fall outside the valid range, further degrading its reliability in critical systems Kingma & Dhariwal (2018).

On the other hand, analytical methods like DipDNN (Yuan et al., 2024) ensure strict one-to-one invertibility at every layer through architectural constraints Dinh et al. (2014). These models are particularly well-suited for applications demanding high precision and exact inversions, such as real-time control and physics simulations. However, this architectural rigidity comes at a cost. DipDNN often underperforms in scenarios where flexible feature extraction—especially through convolutional layers—is necessary, such as in complex image processing tasks. The lack of convolutional layers, which are key for hierarchical feature learning, limits DipDNN's expressiveness and ability to handle high-dimensional, intricate data structures.

Given the complementary nature of iterative and analytical inverse methods, we argue that a single model may not perform optimally across all scenarios. Our extensive analysis of i-ResNet and

Figure 1: Contrast and Complement of i-ResNet's iterative inverse approximation and DipDNN's closed-form inverse. **MetaInv.** A switching algorithm to adapt model on diverse inverse problems.

DipDNN reveals that each method excels in different tasks but also suffers from critical shortcomings in others. Motivated by this observation, we propose a novel meta-inverse (**MetaInv** in Fig. 1) framework that dynamically switches between i-ResNet and DipDNN based on task-specific requirements. This approach leverages the strengths of both architectures—using i-ResNet for tasks requiring flexibility and feature extraction, while relying on DipDNN for tasks demanding strict, one-to-one invertibility. By incorporating a task-driven signal to select between models during inference, the framework adapts to the demands of different applications, ensuring robustness, precision, and computational efficiency.

Our meta-inverse framework provides a more generalized solution for a wide range of inverse learning tasks. The flexibility and adaptability of our approach allow for broader applicability, making it particularly useful in real-world scenarios that require both invertibility guarantees and high performance in feature-rich domains. In this paper, we present a thorough evaluation of our framework across various benchmarks, demonstrating that it significantly outperforms both i-ResNet and DipDNN individually. The results underscore the utility of combining iterative and analytical methods to tackle the diverse challenges faced in inverse learning.

## 2 RELATED WORK

**Invertible Neural Networks (INNs)**. Invertible neural networks (INNs) have emerged as essential tools for a variety of applications that require consistent bidirectional inferences, such as inverse problems, density estimation, and generative modeling. One of the earliest models, NICE Dinh et al. (2014), proposed a coupling layer design that ensures exact invertibility, followed by extensions like RealNVP Dinh et al. (2016), which allowed more expressive feature transformation through affine coupling. Glow Kingma & Dhariwal (2018) further improved these architectures by incorporating 1x1 convolutional layers, making it more suitable for high-dimensional image tasks. Another prominent line of research is iResNet Behrmann et al. (2019), which introduces iterative inversion based on residual blocks, providing flexibility for complex mappings. However, these models have limitations in handling strict one-to-one mappings and suffer from numerical instabilities when dealing with non-contractive mappings.

**Analytical vs. Iterative Inversion Methods**. In the context of inverse problems, there are two main categories of methods: iterative and analytical. Iterative methods like iResNet Behrmann et al. (2019) rely on residual connections and iterative approximation to compute the inverse. While this approach is highly flexible and well-suited for complex tasks, it often struggles with stability and may produce inaccurate inverse mappings, particularly in scenarios where strict one-to-one correspondences are required. On the other hand, analytical inversion methods, such as DipDNN Dupont et al. (2019), enforce strict bijection through architectural constraints. This guarantees invertibility but at the cost of flexibility, particularly in tasks requiring convolutional feature extraction. Analytical approaches also suffer from reduced performance in tasks that require high-dimensional feature extraction, such as image generation and classification.

**Meta-Learning and Hybrid Architectures**. Meta-learning has been explored as a way to dynamically adapt neural network architectures based on the task requirements. Recent works in online control Christianson et al. (2023b) and hybrid models Blum & Burch (1997b) have demonstrated the potential of combining model-based and learning-based approaches. This inspired our hybrid meta-inverse framework, which leverages the strengths of both iterative and analytical methods. Specifically, our method dynamically switches between iResNet and DipDNN layers based on task-specific cues, improving performance in diverse inverse learning tasks. This is similar in spirit to works such as AdaNet Cortes et al. (2017) that adaptively grow neural architectures based on performance metrics, but our approach focuses on invertibility and bidirectional consistency.

**Applications in Real-Time Systems and Inverse Problems**. The need for accurate and efficient inverse mappings is especially pronounced in applications such as scientific computing Raissi et al. (2019), real-time control Levine et al. (2016), and robotics Ziebart et al. (2021). These domains require models that can ensure robustness, precision, and real-time performance, particularly when working with non-contractive or non-bijective mappings. While iterative methods like iResNet provide flexibility, they often fail in applications with strict requirements for inverse consistency and accuracy, such as fluid dynamics and physical system identification. Conversely, analytical methods like DipDNN excel in these scenarios by providing exact inversions, although they lack the flexibility required for more complex tasks like image generation or tasks that involve redundant information. Our meta-inverse framework addresses these limitations by automatically selecting the best method based on the problem requirements, leading to improved robustness and efficiency in real-time applications.

## 3 BACKGROUND: INVERSE PROBLEM AND INVERTIBLE MODELS

### 3.1 PROBLEM STATEMENT AND DISTINCT DEMANDS

Inverse problems span numerous applications that involve recovering original variables from observed outputs This work focuses on inverse learning through invertible mapping recovery, which is for point estimates of images or physical system states. The problem is typically formulated as approximating a forward mapping $f_\theta : \mathbb{R}^n \to \mathbb{R}^n$, where $\boldsymbol{y} = f_\theta(\boldsymbol{x})$ is invertible. The goal is to find the relative inverse mapping $g_\vartheta$ such that $\boldsymbol{x} = g_\vartheta(\boldsymbol{y}) = f_\theta^{-1}(\boldsymbol{y})$, ensuring consistency with the forward process. The demands for such two-way mapping rule recovery are distinct and varied even in one task. Thus, this work evaluates them using the following performance metrics.

- **Forward Prediction Error (Fwd)**: Same as common one-way learning, it measures the ability of the model to predict $\boldsymbol{y}$ from $\boldsymbol{x}$. The invertible model is trained by minimizing the forward prediction loss (any discriminative learning loss $\ell_{\text{fwd}}$):

$$f^* = \operatorname*{argmin}_{\theta} \ell_{\text{fwd}} (\boldsymbol{y}, f_\theta(\boldsymbol{x})) .$$

- **Inverse Reconstruction Error (Inv)**: The reconstruction error evaluates the model's invertibility to recover the original inputs ($\ell_{\text{inv}}$ is mean square error for point estimates),

$$\ell_{\text{inv}} (\boldsymbol{x}, g_\vartheta (f_\theta(\boldsymbol{x}))) .$$

- **Inverse Consistency (Inv-Consist)**: It assesses the consistency between forward and inverse mappings by comparing the inverse predictions with the true labels $\boldsymbol{y}$, rather than just the forward outputs:

$$\ell_{\text{fwd}} (\boldsymbol{x}, g_\theta(\boldsymbol{y})) .$$

It is important to note the distinction between inverse accuracy and consistency. Inverse accuracy, or reconstruction error, indicates only the model's invertibility to compute the inverse, where analytical invertibility yields an error-free result, and numerical invertibility minimizes round-off errors. Consistency, on the other hand, evaluates the model's capability of two-way learning. It reveals the precision of global inversion, e.g., minimizing reconstruction error doesn't necessarily need a low forward approximation error, while the consistency error integrates errors from both forward and inverse processes. Most works involving approximate one-to-one mappings focus on inverse accuracy alone, such as the image classification and recovery (Behrmann et al., 2019).

## 3.2 Review of Inverse Learning Models

Existing invertible neural networks (INNs) can be broadly categorized into two approaches: iterative numerical approximations and analytical inverse solutions.

**Iterative Numerical Methods:** I-ResNet (Behrmann et al., 2019) and related models compute inverse mappings iteratively. I-ResNet guarantees numerical stability in inversion by designing residual blocks $f_{\text{i-ResNet}}^{(k)}(\boldsymbol{z}^k) = \boldsymbol{z}^{k+1} = \boldsymbol{z}^k + h^{(k)}(\boldsymbol{z}^k)$, where $h^{(k)}$ is a nonlinear function that satisfies $\text{Lip}(h^{(k)}) < 1$ to ensure a contractive transformation. While this approach is computationally flexible, the inverse $\boldsymbol{z}^k = \left(f^{(k)}\right)^{-1}(\boldsymbol{z}^{k+1})$ is obtained through iterative refinement, which can be inefficient and prone to convergence issues, especially in scenarios where the Lipschitz condition fails to hold.

**Analytical Inverse Methods:** These methods aim to provide closed-form inverses through algebraic constructions. For instance, NICE (Dinh et al., 2014) and Glow (Kingma & Dhariwal, 2018) achieve bijective transformations by using coupling layers that split input and output variables. The forward transformation is given by: $\boldsymbol{z}_1^{k+1} = a\boldsymbol{z}_1^k$, $\boldsymbol{z}_2^{k+1} = b\boldsymbol{z}_2^k + t(\boldsymbol{z}_1^k)$, and the inverse is computed as: $\boldsymbol{z}_1^k = \frac{\boldsymbol{z}_1^{k+1}}{a}$, $\boldsymbol{z}_2^k = \frac{\boldsymbol{z}_2^{k+1} - t(\boldsymbol{z}_1^k)}{b}$. Here, $a, b \neq 0$ are constants, and $t(\cdot)$ is a nonlinear function (e.g., NN). While this ensures an analytical inverse, it imposes significant structural constraints that can limit expressiveness, especially in complex, high-dimensional tasks such as feature extraction and image modeling.

**Decomposed Invertible Pathway DNN (DipDNN):** DipDNN (Yuan et al., 2024) avoids strict partitioning by introducing a layer-wise decomposition with triangular weight matrices. The forward mapping in DipDNN is expressed as:

$$f_{\text{DipDNN}}^{(k)}(\boldsymbol{z}^k) = g_2^{(k)}(W_{\text{tri}}^{(k)} g_1^{(k)}(W_{\text{tril}}^{(k)} \boldsymbol{z}^k + b_1^{(k)}) + b_2^{(k)}).$$

The triangular structure of $W_{\text{tril}}$ and $W_{\text{tri}}$ ensures strict bijection, while the use of monotonic activation functions such as LeakyReLU or ELU preserves expressiveness. DipDNN offers a more flexible architecture that avoids the constrained coupling layers of NICE and Glow, enabling efficient inversions without sacrificing approximation power.

## 4 Invertibility vs. Approximation Limitations of i-ResNet

I-ResNet and its variants have achieved substantial success in tasks such as image classification and density estimation by addressing the challenge of invertibility. However, these models struggle with the trade-off between expressive power and the need for stable inversion, imposed by the Lipschitz condition. This constraint forces each residual block to make minimal adjustments to the input, limiting the model's ability to approximate complex mappings that involve significant changes or intricate dependencies (Zhang et al., 2020b). This section provides theoretical analysis to explore these limitations in depth.

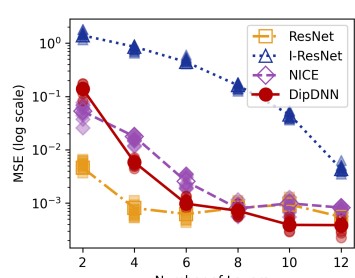

Figure 2: Fwd approximation capability on synthetic example.

### 4.1 Contrasting Invertibility Between i-ResNet and DipDNN

**Layer-Wise Contraction and Limited Scaling.** The following theorem formalizes the limitation of i-ResNet in terms of its ability to approximate functions with high Lipschitz constants:

**Theorem 1.** *For a $K$-layer i-ResNet, where each residual block $h^{(k)}$ satisfies the Lipschitz constraint $\text{Lip}(h^{(k)}) < 1$. Let $f_T : \mathbb{R}^d \to \mathbb{R}^d$ be a multi-dimensional mapping that may be nonlinear, with a Lipschitz constant $\text{Lip}(f_T) = L_T > 1$. Then, a $K$-layer i-ResNet cannot adequately approximate $f_T$ if $L_T > 2^K$, and the relative approximation error is bounded by: $\epsilon = \Omega\left(1 - \frac{2^K}{L_T}\right)$.*

The proof are included in Appendix A.1 and A.2. This result indicates that as the complexity of the target function increases (i.e., as $L_T$ grows), i-ResNet's ability to represent it deteriorates unless the number of layers is increased accordingly. For real-world applications, the number of layers required can be approximated as:

$$K_T \geq \frac{\log(\hat{K})}{\log(1 + \text{Lip}(f_{\text{i-ResNet}}))},$$

where $\hat{K} = \max_i \frac{\max(\boldsymbol{y}_i) - \min(\boldsymbol{y}_i)}{\max(\boldsymbol{x}_i) - \min(\boldsymbol{x}_i)}$. Detailed derivations of this estimate can be found in Appendix A.2. Empirical evidence suggests that adaptive layer estimation is effective when evaluated on small data batches.

**Inadequacy for Sign-Flipping.** While deeper architectures can mitigate the limitation of layer-wise contraction, i-ResNet faces an additional challenge in modeling mappings that involve sign-flipping in the inputs. The following theorem establishes that i-ResNet cannot approximate functions where input and output vectors have opposite signs. The corresponding proof is included in Appendix A.3.

**Theorem 2.** *For a $K$-layer i-ResNet, where each residual block $h^{(k)}$ satisfies the Lipschitz constraint $Lip(h^{(k)}) < 1$, let $f_T : \mathbb{R}^d \to \mathbb{R}^d$ be a multi-dimensional mapping such that $\boldsymbol{x} \cdot f_T(\boldsymbol{x}) < 0$ (i.e., the signs of $\boldsymbol{x}$ and $f_T(\boldsymbol{x})$ are opposite in at least one dimension). We claim that i-ResNet $f_{\text{i-ResNet}}$ cannot approximate such a mapping $f_T$.*

**Non-Strict One-to-One Mapping Approximation via Iterative Inverse.** The layer-wise contraction of i-ResNet limits the degree to which input space can be scaled or bent during forward transformations. However, this approach to enforcing invertibility provides flexibility in the network's architecture, e.g., convolution layers within the residual blocks. Specifically, the convolution operation is non-invertible due to its many-to-one mapping:

$y[i] = (\boldsymbol{x} * \boldsymbol{w})[i] = \sum_{j=1}^{k} w[j] \cdot x[i - j],$

| Model | Config. | Acc. |
|-------|---------|------|
| ResNet | Conv | 97.89% |
| | MLP | 97.65% |
| I-ResNet | Conv | 96.32% |
| | MLP | 97.20% |
| DipDNN | MLP | 97.69% |

(a)

| Model | $k = \{1, 1, 1\}$ | $k = \{3, 1, 3\}$ |
|-------|-------|-------|
| ResNet - Conv | 88.06% | 90.79% |
| I-ResNet - Conv | 83.22% | 90.26% |
| DipDNN - Conv | 84.29% | 90.69% |
| ResNet - MLP | 66.97% | - |
| I-ResNet - MLP | 66.69% | - |

(b)

| | I-ResNet | I-ResNet $k = 1$ | I-ResNet $k = 2$ | DipDNN |
|-------|-------|-------|-------|-------|
| Fluid Dynamics | 0.1635 | 7.7005 | 0.3343 | 0.1258 |
| Power Flow | 5.73e-05 | 2.08e-05 | 6.95e-05 | 2.01e-06 |

(c)

Figure 3: (a) & (b) MNIST & CIFAR-10 classification accuracy; (c) Physical system modeling performance.

where $\boldsymbol{w}$ is the convolution filter or kernel. With padding, convolution aims to extract features from redundant information in the input, which is advantageous for tasks like classification involving high-dimensional images. In particular, convolution helps distill features and reduce dimensionality, improving performance (*e.g.*, Fig. 3 shows improved accuracy due to dimensionality reduction).

Nonetheless, there are many inverse problems where the redundancy of the information is minimal, such as in fluid dynamics, sonar sensor assimilation, and power flow analysis. In these cases, either the system states (spatial position, time) or physical properties (velocity, pressure, etc.) are loosely dependent on one another. Importantly, physical systems typically have explicit forward functions, and the ability to produce accurate point estimates for modeling and control is more critical than generating qualitative results such as images. In fluid flow modeling, for instance, Navier-Stokes equations govern the system, and any loss of information leads to ambiguous results or reconstruction errors. Under such scenarios, as Fig. 3 (c) indicates, i-ResNet's iterative approximation of non-strict one-to-one mappings may result in inaccuracies and lack of exactness in the inverse computations.

**Layer-Wise Bijective Transformation for Analytical Inverse.** In contrast to the iterative nature of i-ResNet, DipDNN enforces invertibility through layer-wise one-to-one transformations. The proposition to guarantee bijective layer-wise transformation is in Appendix A.4. Unlike i-ResNet's Lipschitz constraints, which limit flexibility, DipDNN imposes no limitations on the network's ability to scale features or flip the signs of inputs.

**Equivalence & Conflict Between i-ResNet and DipDNN.** While the strict bijective transformations of DipDNN provide guaranteed invertibility, this structure can be overly restrictive for some inverse problems where approximate inverses suffice. i-ResNet, with its flexibility (e.g., convolution layers), may outperform DipDNN in these cases. This raises an important question: can DipDNN achieve similar performance for these tasks?

To address this, we aim to show an equivalence between convolution-based residual networks and DipDNN based on fully connected layers, for which the proof is in Appendix A.5.

**Theorem 3.** *Consider a $K$-layer convolution-based i-ResNet. Let each convolution layer $h_i^{(k)}(z^k) = g(W_{conv}^{(k)} * z^k + b)$, where $W_{conv} \in \mathbb{R}^{k \times k}$. Then:*

1) *Each residual block $h_i^{(k)}$ can be represented in an equivalent form in DipDNN, satisfying Proposition 1.*

2) *The residual connection $f_{i\text{-}ResNet}^{(k)}(z^k) = z^k + h^{(k)}(z^k)$ inherently violates the strict invertibility of DipDNN.*

## 4.2 META-INVERSE ALGORITHM FOR COMPLEMENTARY SELECTION

From Sections 4 and 4.1, we observe the contrasting strengths and weaknesses of i-ResNet and DipDNN. i-ResNet excels in providing flexible architectures, such as non-bijective convolutional feature extraction, but struggles to maintain strict one-to-one mappings and compensates poorly when faced with contraction limitations. On the other hand, DipDNN guarantees analytical invertibility through its structured, layer-wise design, but this strictness can be overly limiting for inverse problems that exhibit redundancy in the data. These differences highlight the need for a hybrid approach capable of dynamically adapting to task-specific characteristics, e.g., complexity, redundancy, and precision requirements of inverse problems.

To address this, we propose the Meta-Inverse (**MetaInv**) algorithm, which dynamically switches between i-ResNet and DipDNN based on the specific characteristics of the task at hand. The MetaInv algorithm in Fig. 4 is inspired by learning-augmented switching methods in online control (Christianson et al., 2023a; Blum & Burch, 1997a), where an agent alternates between a learning-based approach with high expressive power (i-ResNet) and a model-based approach with lower risk (DipDNN). Specifically, as outlined in Section 3.1, the MetaInv algorithm is designed to balance four key requirements of inverse learning:

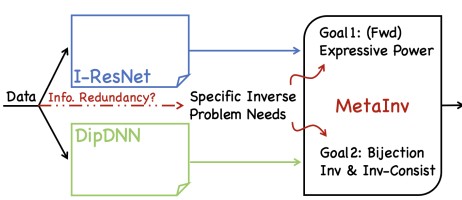
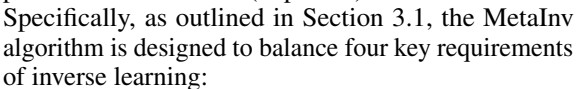

Figure 4: Meta-Inverse Algorithm: Switching between i-ResNet and DipDNN based on task characteristics.

(a) **Forward expressive power**: Modeling complex mappings correctly with high fidelity.

(b) **Tight error bounds on the inverse**: Minimizing inverse reconstruction error, ideally achieving values close to zero (theoretically) or minimizing numerical round-off errors.

(c) **Consistency of bi-directional mappings**: Ensuring that both forward and inverse mappings are consistent when true labels are used for inverse predictions.

(d) **Computational efficiency**: Optimizing resource usage to ensure that the model is both computationally efficient and scalable to large datasets.

**Trust-Weighted Switching Mechanism:** The core of the MetaInv algorithm (Algorithm 2) is a trust-weighted switching mechanism that integrates task-specific performance metrics (Fwd, Inv, Inv-Consist) defined in Sec. 3.1 with computational cost into a single evaluation function: $V_{\text{model}} = J_{\text{model, total}} + \lambda C_{\text{model}}$, where $J_{\text{model, total}}$ is the weighted combination of performance metrics, $\lambda$ is the weight for computational cost, and $C_{\text{model}}$ includes time complexity and model size.

To ensure stability and prevent trivial convergence, the trust-weight parameter $\beta$ is updated iteratively based on the performance difference between i-ResNet and DipDNN: $\beta_{t+1} = \beta_t + \eta_t(V_{\text{i-ResNet}} - V_{\text{DipDNN}})$, where $\eta_t$ is the learning rate (empirically set to 0.01). Additionally, $J_{\text{threshold}}$ sets an acceptable range of performance, preventing the algorithm from stagnating when inverse learning is sensitive to noise or data uncertainties. This mechanism allows the algorithm to smoothly adapt to the optimal model for an arbitrary task.

The evaluation of task-specific metrics is not fixed and evolves dynamically during learning dynamically switches between the two architectures, optimizing both forward and inverse computation. This

---

**Algorithm 1** Meta-Inverse Algorithm Between i-ResNet and DipDNN

---

**Input:** $\{\boldsymbol{x}_i, \boldsymbol{y}_i\}_{i=1}^N$ (data), $J_{\text{threshold}}$ (performance threshold), $\lambda$ (weights for performance and cost), $T$ (iterations)
**Output:** Optimal selection of i-ResNet or DipDNN for computations
Initialize $\beta_0, \lambda$
**for** $t = 1, \ldots, T$ **do**
    **Forward and Inverse Computation**
    Compute forward, inverse (reconstruction and ablation) outputs: $\boldsymbol{y}_{\text{model}}, \hat{\boldsymbol{x}}_{\text{model}}, \hat{\boldsymbol{x}}'_{\text{model}}$ for model $\in \{$i-ResNet, DipDNN$\}$,
    **Metric Evaluation**
    Calculate **Fwd Acc**, **Inv Acc**, and **Inv Consist** losses and compute combined scores: $J_{\text{model, total}} = \sum_{k=1}^3 \lambda_k J_{\text{model, k}}$, where $J_{\text{model, k}}$ represents the respective metrics for $k = 1, 2, 3$.
    **Model Selection**
    **if** $J_{\text{DipDNN, total}} < J_{\text{i-ResNet, total}}$ **and** $J_{\text{DipDNN, total}} < J_{\text{threshold}}$ **then**
        Use DipDNN: $\boldsymbol{y} = \boldsymbol{y}_{\text{DipDNN}}, \boldsymbol{x}_{\text{reconstructed}} = \boldsymbol{x}_{\text{DipDNN}}$
    **else**
        Use i-ResNet: $\boldsymbol{y} = \boldsymbol{y}_{\text{i-ResNet}}, \boldsymbol{x}_{\text{reconstructed}} = \boldsymbol{x}_{\text{i-ResNet}}$
    **end if**
    **Trust-Weight Update**
    Compute computational cost $C_{\text{model}}$ for both models: $C_{\text{model}} = \alpha_1 T_{\text{model}} + \alpha_2 M_{\text{model}} + \alpha_3 S_{\text{model}}$
    Compute total evaluation cost $V_{\text{model}}$ for each model: $V_{\text{model}} = J_{\text{model, total}} + \lambda C_{\text{model}}$
    Update trust-weight $\beta_{t+1} = \beta_t + \eta_t(V_{\text{i-ResNet}} - V_{\text{DipDNN}})$
**end for**

---

approach corrects the limitations of prior evaluations that were either incomplete or misaligned with the actual data characteristics and task-specific requirements. Details of the MetaInv algorithm's implementation and evaluation can be found in the Appendix.

# 5 NUMERICAL EXPERIMENTS

Throughout experiments on diverse inverse problems, we aim to answer the following questions: 1) For non-contractive cases (scaling & sign-flipping), can DipDNN outperform i-ResNet on bidirectional mappings? 2) Iterative inverse approximation vs. analytical inverse on non-strict bijective cases of different information redundancy, which wins out? 3) Can MetaInv switch to the optimal model for arbitrary Inverse problems? To answer these questions, we perform our experiments on different inverse problems: synthetic examples, image classification (Fwd) and reconstruction (Inv) from last features (Behrmann et al., 2019), image transformation with inverted colors for reversible data hiding (Zhang et al., 2024), power flow modeling (Fwd) and state estimation (Inv), and Navier-Stokes flow dynamics modeling (Fwd) and initial condition inference (Inv) (Langtangen & Logg, 2017). Each task represents important application needs in inverse problems.

For comparison, we test 1) **i-ResNet**, which is representative of iterative inverse approximation; 2) **ResNet**, a baseline to reveal the standard approximation capability compared with contractive i-ResNet; 3) **NICE**, a baseline analytical inverse model with strict architecture constraint; 4) **DipDNN**, an improved analytical inverse with layer-wise bijective transformation. Finally, we compare them with the **MetaInv**, the proposed switching algorithm to adaptively select between i-ResNet and DipDNN. Note that we select the most representative invertible NN structures, whose variants have not shown significant differences for the targeted broad applications in both image-related and physical system tasks. For example, although Glow (based on NICE) is not directly tested, we implement its invertible ActNorm and $1 \times 1$ convolution in experiments. For all the testing scenarios, all the inverse learning models contain similar numbers of parameters in each comparison for a fair evaluation. For each inverse learning method, we equally evaluate the performance using the error metrics of forward prediction, inverse reconstruction, and inverse consistency, as defined in Sec. 3.1. Specially for image tasks, we show qualitative results for intuitive visualization.

## 5.1 INVERTIBILITY VS. EXPRESSIVE POWER ON NON-CONTRACTIVE CASES

Based on the theoretical analysis in Sec. 4, i-ResNet imposes limitations in expressive power. We start with synthetic datasets to quantify the limitations. Further, we conduct sensitivity analysis and visualize qualitative results to understand the trade-off between invertibility and approximation capability.

**Sensitivity Analysis of Invertible NN Depth.** To test the number of layers needed for i-ResNet, we evaluate the performance of synthetic data (Fig. 2) and power flow cases (Fig. 5), which exhibit non-contractive mappings. It's obvious that while DipDNN approaches a similar fwd error as ResNet, i-ResNet has a much higher error. It requires a deeper model to reach the same performance as others.

**Color Inversion Mixed with Complex Transformation.**
The invertibility enforced by contraction imposes the inability to approximate $x f(x) < 0$. Except for the synthetic data with the flipped sign of inputs and outputs (Fig. 10), we generalize such a property to more complex image examples. It is a reversible data-hiding task that mixes color inversion (e.g., white to black and black to white) and class transformation (e.g., one item to another) to cover the privacy information of original images Fig. 6. The data-hiding process needs to be lossless to recover back to the original data.

In the synthetic example with flipped signs, Fig. 10 in the Appendix shows that the error of i-ResNet cannot decrease no matter how many more layers are used. Other invertible NNs successfully capture this relationship but NICE cannot maintain the inverse performance on both reconstruction (middle column) and independent prediction (last column).

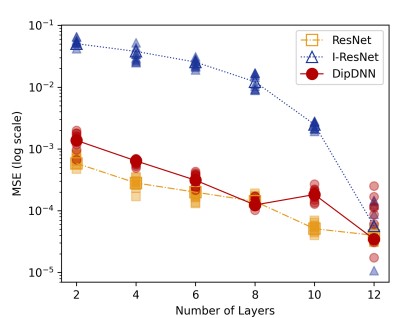

Figure 5: Fwd approximation capability on power flow example.

In a similar context, we construct the data-hiding case in images (Fig. 6) using images from MNIST, FashionMNIST, and CIFAR-10. Regular images are used as inputs; we create outputs by combining half of the color-converted MNIST images with half of the FashionMNIST images and, alternatively, a quarter of color-converted MNIST images with three-quarters of FashionMNIST images. When gradually mixing image class transformation and color inversion (a portion of $\{0, 1/2, 1/4\}$), i-ResNet is hard to find the correct mapping. Specifically, the 3rd and 4th rows in Fig. 6 indicate that i-ResNet merely learns. Analytical invertible NNs like NICE and DipDNN not only have no restrictions but also learn the nonlinear transformation well. Moreover, DipDNN has superior inverse consistency for balanced performances of bi-directional mappings.

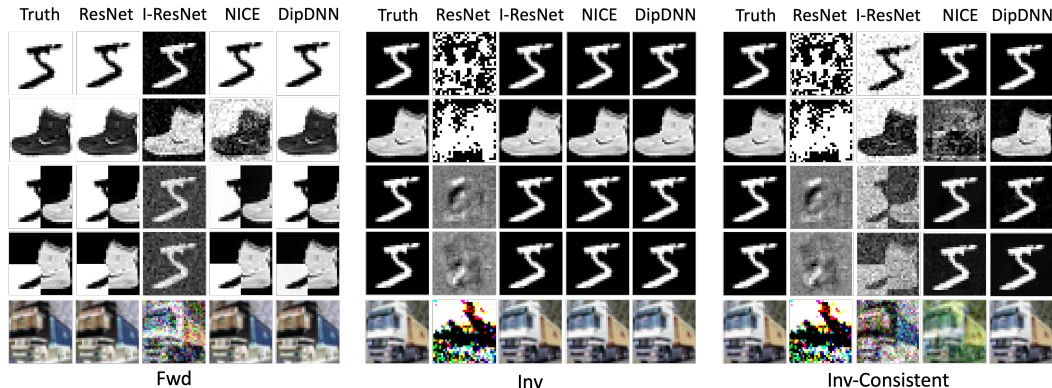

Figure 6: Performance of different models on color inversion mixed with complex transformation.

## 5.2 Approximated Inverse vs. Closed-Form Inverse on different information redundancy

In Sec. 4.1, we analyze that the performances of invertible NNs depend on the data and problem need. Thus, we test on two representative physical and engineering systems other than the previous image-related tasks. Unlike the image classification case, the data of power system measurements and fluid dynamics observations have limited information redundancy as they are clearly defined system states or conditions.

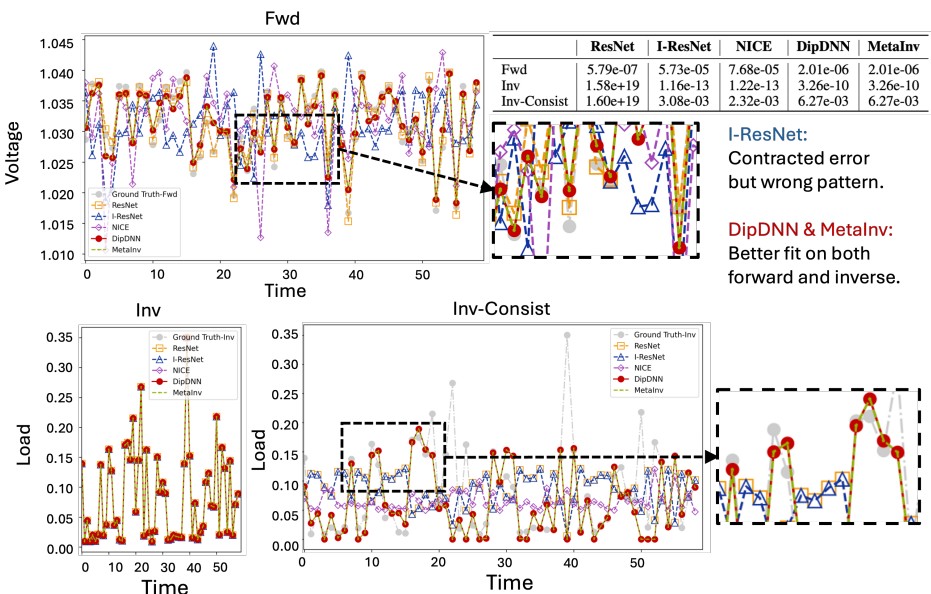

Figure 7: Performance comparison of different methods on the power flow case. Top-left: forward voltage predictions (Fwd); Bottom plots: inverse load recovery (Inv and Inv-Consist); Top-right Table: numerical error metrics. I-ResNet achieves tightly bounded errors but predicts deviate from the ground truth. DipDNN captures system dynamics more effectively, which MetaInv selected to balance predictive accuracy and inverse alignment.

As for the power flow case, the data is generated from the simulation of grid power flow equations, which govern the system-wide power injections based on voltage phasors and the system's topology and parameters. The power flow dataset is built on the load inputs to estimate the voltage at different buses in an 8-bus system for forward mapping. In this case, inverse learning is essential to recover the underlying system physics and estimate load conditions for better interpretability and consistency in power flow analysis.

The power system experiment shows how different methods predict voltage (Fwd) and load (Inv) under limited measurements Fig. 7. DipDNN and MetaInv outperform i-ResNet using the iterative inverse. Although their errors are similar in the table at the top-right corner, DipDNN fits individual points much better. This is because i-ResNet has the contractive property enforced by Lip $< 1$: even though the point estimates are bad, the average error is bounded tightly. DipDNN is selected by MetaInv for the power flow case. The voltage plot (Fwd) reveals that I-ResNet, despite contracting the error, produces a pattern that does not align with the ground truth. Conversely, DipDNN achieves better consistency and captures the essential dynamics of the system, especially in the inverse problem (Inv-Consist). This demonstrates how critical it is for the method to maintain a balance between predictive power and inverse accuracy. Moreover, convolutional layers, while useful for forward problems, might introduce excessive contraction in systems with low redundancy, leading to distortions in the inverse process.

As for the fluid dynamics case, the data is generated from the simulation of a chemical reaction between species A and B, which form species C, while considering advection and diffusion in a domain governed by a coupled system of partial differential equations. The dataset is built on the

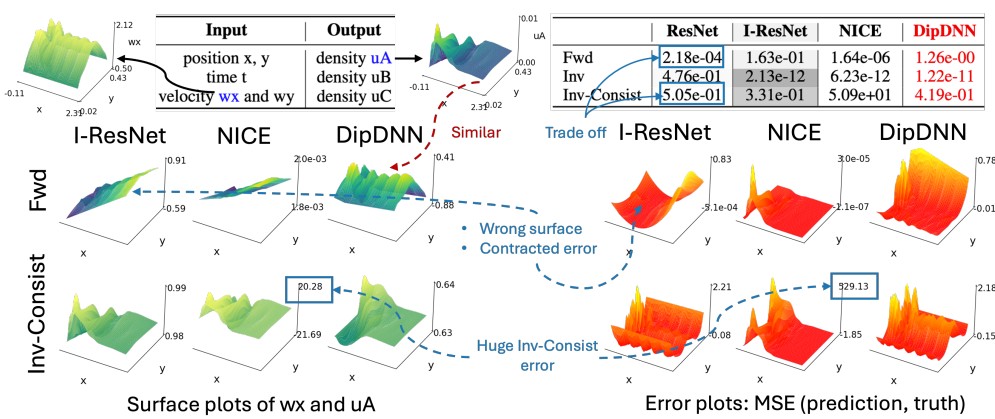

Figure 8: Forward and inverse performance of different models on Navier-Stokes flow dynamics.

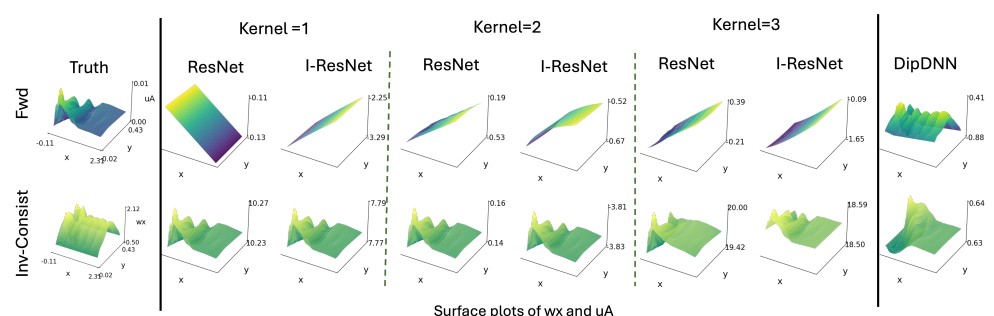

Figure 9: Performance of convolutional models on Navier-Stokes flow dynamics.

concentrations of these species, along with the velocity field, to capture the spatio-temporal evolution of the chemical reaction and the dynamics of fluid flow. In this case, inverse learning is needed to recover system dynamics and parameters from observed concentration data for interpretability and consistency. With dense observation data, Fig. 8 depicts the surface plots for forward (Fwd) and inverse consistency (Inv-Consist) of the velocity fields (wx and uA) for multiple models. DipDNN shows a superior fit for both forward and inverse consistency, especially in cases where the information redundancy is higher, and the system is more over-determined. For i-ResNet, although it attempts to approximate the inverse, its surface predictions deviate significantly, reflecting its struggle with preserving the intricate dynamics of the fluid flow.

The error plots comparing prediction and truth further reinforce this. DipDNN maintains a low mean squared error (MSE) across both forward and inverse mappings, while i-ResNet struggles with a high inverse consistency error. This discrepancy highlights the limitation of iterative inverse approximation, especially when the system's redundant information is minimal. In this case, DipDNN's layer-wise bijective transformations enable a more robust handling of both the advection and diffusion terms in the PDEs, capturing the fluid dynamics more precisely.

Figure 9 provides insights into the impact of convolutional layers when information redundancy is low. As seen in the forward and inverse consistency surface plots, both ResNet and I-ResNet produce distorted surfaces. Convolutional layers introduce locality in modeling, which is beneficial with high information redundancy. However, in systems like Navier-Stokes dynamics with limited information, the imposed locality leads to contracted errors and incorrect surfaces. The experiments show that DipDNN, which avoids heavy convolutional operations, performs significantly better in such low-redundancy cases.

# 6 CONCLUSION

In this paper, we analyze the complementary strengths and inherent limitations of typical invertible models DipDNN and i-ResNet, highlighting the challenges of applying either model individually across a wide range of inverse problems. DipDNN ensures strict analytical inversion with precise one-to-one mapping but lacks the flexibility to incorporate convolutional layers for feature extraction, whereas i-ResNet effectively integrates convolution with residual connections, enhancing feature extraction and scalability; however, its Lipschitz constraints limit forward approximation, compromising inverse accuracy in non-contractive scenarios. Thus, we propose a meta-inverse algorithm to leverage their respective strengths on specific inverse problems. MetaInv balances three critical metrics—forward accuracy, inverse consistency, and inverse accuracy—addressing the shortcomings of prior work that focused on a subset of these metrics. By coupling i-ResNet and DipDNN effectively, our solution ensures a more comprehensive evaluation and a fairer comparison for diverse inverse problems.

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

## A  APPENDIX

### A.1  PROOF OF THEOREM 1

*Proof.* Given that $\text{Lip}(h^{(k)}) < 1$, the Lipschitz constant of each residual layer satisfies $\text{Lip}(f^{(k)}) < 2$. Hence, for a $K$-layer i-ResNet, the overall Lipschitz constant of the network is bounded by:

$$\text{Lip}(f) = \prod_{k=1}^{K}(1 + \text{Lip}(h^{(k)})) < 2^K.$$

If the target function $f_T$ has a Lipschitz constant $L_T > 2^K$, the i-ResNet is unable to approximate it accurately since its representation capacity is limited by $2^K$. The relative approximation error across $\epsilon$ dimensions is thus bounded by:

$$\epsilon = \frac{\|f(\boldsymbol{x}) - f_T(\boldsymbol{x})\|}{\|f_T(\boldsymbol{x})\|} = \Omega\left(1 - \frac{2^K}{L_T}\right).$$

$\square$

### A.2  ESTIMATING THE NUMBER OF LAYERS NEEDED FOR I-RESNET

To approximate a target mapping $f_T$ with $f_{\text{i-ResNet}}$, the effective Lipschitz constant $\hat{K}$ can be empirically estimated from the data as:

$$\hat{K} = \max_i V_i, \quad V_i = \frac{\max(y_i) - \min(y_i)}{\max(x_i) - \min(x_i)}, \quad i = 1, \ldots, d.$$

For an i-ResNet with $K$ layers, the effective Lipschitz constant can be approximated as:

$$\text{Lip}(f_{\text{i-ResNet}}) \approx (1 + \text{Lip}(h^{(k)}))^K.$$

Note that $\text{Lip}(h^{(k)})$ can be adjusted close to a number $< 1$ in experiments and we use it for estimation. To approximate $f_T(\boldsymbol{x})$, the number of layers $K_T$ necessarily needed is estimated by:

$$K_T \geq \frac{\log(\hat{K})}{\log(1 + \text{Lip}(h^{(k)}))}.$$

If $\hat{K} \gg 1$, then $K_T$ has to be sufficiently large for forward accuracy, compared to ResNet (without constraints) and DipDNN.

### A.3 SUPPLEMENTARY PROOF OF FLIP-SIGN MAPPING FOR I-RESNET IN THEOREM 2

*Proof.* Consider a target mapping $f_T(\boldsymbol{x})$ where the dot product $\boldsymbol{x} \cdot f_T(\boldsymbol{x}) < 0$, indicating that the inputs and outputs have opposite signs in at least one dimension. To demonstrate that i-ResNet cannot model this mapping, we show that for any $\boldsymbol{x}$, $\boldsymbol{x} \cdot f_{\text{i-ResNet}}(\boldsymbol{x}) \geq 0$ holds.

From (Zhang et al., 2020b), consider two inputs $\boldsymbol{x}_1^0$ and $\boldsymbol{x}_2^0 = \boldsymbol{x}_1^0 + \delta_0$. After one residual block, the outputs are $\boldsymbol{x}_1^1 = \boldsymbol{x}_1^0 + h^{(1)}(\boldsymbol{x}_1^0)$ and $\boldsymbol{x}_2^1 = \boldsymbol{x}_1^0 + \delta_0 + h^{(1)}(\boldsymbol{x}_1^0 + \delta_0)$. The difference between $\boldsymbol{x}_1^1$ and $\boldsymbol{x}_2^1$ is given by:

$$\delta_1 = \boldsymbol{x}_2^1 - \boldsymbol{x}_1^1 = \delta_0 + \left( h^{(1)}(\boldsymbol{x}_1^0 + \delta_0) - h^{(1)}(\boldsymbol{x}_1^0) \right).$$

Since $\text{Lip}(h^{(1)}) < 1$, it follows that:

$$|\delta_1| < |\delta_0|,$$

implying that the transformation preserves the sign relationships between $\boldsymbol{x}_1^0$ and $\boldsymbol{x}_2^0$. Applying this logic iteratively to deeper layers shows that for any input $\boldsymbol{x}$, the residual layers maintain $\boldsymbol{x} \cdot f_{\text{i-ResNet}}(\boldsymbol{x}) \geq 0$.

Thus, i-ResNet cannot model mappings where $\boldsymbol{x} \cdot f_T(\boldsymbol{x}) < 0$. The relative approximation error for such mappings is quantified by:

$$\epsilon = \inf_{f_{\text{i-ResNet}}} \frac{\|f_T(\boldsymbol{x}) - f_{\text{i-ResNet}}(\boldsymbol{x})\|}{\|f_T(\boldsymbol{x})\|} \geq \Omega(1),$$

where the error is dominated by the components with flipped signs. □

### A.4 GUARANTEE INVERTIBILITY OF DIPDNN

**Proposition 1.** *The neural network model $f_{DipDNN} : \mathbb{R}^n \to \mathbb{R}^n$, defined as $f_{DipDNN} = f_{DipDNN}^{(1)} \circ \cdots \circ f_{DipDNN}^{(K)}$, is invertible if the weight matrices $W_{tril}^k$ and $W_{triu}^k$, for $k \in [1, K]$, are lower and upper triangular matrices with non-zero diagonal elements. Each block $f_{DipDNN}^{(k)}(\boldsymbol{z}^k)$ is given by:*

$$f_{DipDNN}^{(k)}(\boldsymbol{z}^{(k)}) = g_2^k(W_{tril}^k g_1^k(W_{triu}^k \boldsymbol{z}^k + b_1^k) + b_2^k),$$

*where $W_{tril}^k, W_{triu}^k \in \mathbb{R}^{n \times n}$, and the non-zero diagonal elements of $W_{tril}^k$ and $W_{triu}^k$ ensure invertibility via the triangular structure. The activation functions $g_1^k$ and $g_2^k$ are strictly monotonic, ensuring bijective transformations for each layer.*

The inverse of the DipDNN transformation can be computed in closed form as follows:

$$f_{\text{DipDNN}}^{(k)^{-1}}(\boldsymbol{z}^{k+1}) = \left( W_{\text{triu}}^k \right)^{-1} \left( g_1^k \right)^{-1} \left( \left( W_{\text{tril}}^k \right)^{-1} \left( \left( g_2^k \right)^{-1} \left( \boldsymbol{z}^{k+1} - b_2^k \right) \right) - b_1^k \right).$$

### A.5 PROOF OF THEOREM 3

*Proof.* For *1)*, we rewrite the convolution operation as matrix multiplication, with $T(W_{\text{conv}}^{(k)})$ representing the convolution kernel as a Toeplitz matrix:

$$h_i^{(k)}(\boldsymbol{z}^k) = g(W_{\text{conv}}^{(k)} * \boldsymbol{z}^k + b) = g(T(W_{\text{conv}}^{(k)})\boldsymbol{z}^k + b) = g(L_{\text{conv}}^{(k)} U_{\text{conv}}^{(k)} \boldsymbol{z}^k + b).$$

Applying LU decomposition to the Toeplitz matrix, we obtain lower and upper triangular matrices, ensuring invertibility. Therefore, in DipDNN, we assign $W_{\text{tril}}^k = L_{\text{conv}}^{(k)}$, $W_{\text{triu}}^k = U_{\text{conv}}^{(k)}$, and $g_1^k$ as identity, resulting in:

$$f_{\text{DipDNN}}^{(k)}(\boldsymbol{z}^k) = g_2^k(W_{\text{tril}}^k g_1^k(W_{\text{triu}}^k \boldsymbol{z}^k + b_1^k) + b_2^k) = g(L_{\text{conv}}^{(k)} U_{\text{conv}}^{(k)} \boldsymbol{z}^k) = h_i^{(k)}(\boldsymbol{z}^k).$$

For *2)*, the residual connection $f_{\text{i-ResNet}}^{(k)}(\boldsymbol{z}^k) = \boldsymbol{z}^k + h^{(k)}(\boldsymbol{z}^k)$ introduces an additive identity term that violates DipDNN's strict triangular structure. The addition imposes ambiguity, as the nonlinear dependency between $h_i^{(k)}(\boldsymbol{z})$ and $\boldsymbol{z}$ precludes a closed-form inverse solution. Thus, DipDNN's architecture conflicts with i-ResNet's residual addition, as DipDNN requires strict bijective mappings. □

## A.6 DETAILS OF META-INVERSE ALGORITHM IMPLEMENTATION

We take inspiration from trust-region policy optimization and learning-augmented switching algorithms for selecting control agents. MetaInv aims to dynamically select between i-ResNet (analogous to a learning-based, higher-expressive-power agent) and DipDNN (analogous to a model-based, lower-risk agent) similarly so that the overall system's performance is optimized with stability guarantees.

---

**Algorithm 2** Meta-Inverse Algorithm Between i-ResNet and DipDNN

---

**Input:** $\{\boldsymbol{x}_i, \boldsymbol{y}_i\}_{i=1}^N$ (data), $J_{\text{threshold}}$ (performance threshold), $\lambda$ (weights for performance and cost), $T$ (iterations)
**Output:** Optimal selection of i-ResNet or DipDNN for computations
Initialize $\beta_0, \lambda$
**for** $t = 1, \ldots, T$ **do**
  **Forward and Inverse Computation**
  Compute forward, inverse (reconstruction and ablation) outputs: $\boldsymbol{y}_{\text{model}}, \hat{\boldsymbol{x}}_{\text{model}}, \hat{\boldsymbol{x}}'_{\text{model}}$ for model $\in \{\text{i-ResNet, DipDNN}\}$,
  **Metric Evaluation**
  Calculate **Fwd Acc**, **Inv Acc**, and **Inv Consist** losses and compute combined scores: $J_{\text{model, total}} = \sum_{k=1}^3 \lambda_k J_{\text{model, k}}$, where $J_{\text{model, k}}$ represents the respective metrics for $k = 1, 2, 3$.
  **Model Selection**
  **if** $J_{\text{DipDNN, total}} < J_{\text{i-ResNet, total}}$ **and** $J_{\text{DipDNN, total}} < J_{\text{threshold}}$ **then**
    Use DipDNN: $\boldsymbol{y} = \boldsymbol{y}_{\text{DipDNN}}, \boldsymbol{x}_{\text{reconstructed}} = \boldsymbol{x}_{\text{DipDNN}}$
  **else**
    Use i-ResNet: $\boldsymbol{y} = \boldsymbol{y}_{\text{i-ResNet}}, \boldsymbol{x}_{\text{reconstructed}} = \boldsymbol{x}_{\text{i-ResNet}}$
  **end if**
  **Trust-Weight Update**
  Compute computational cost $C_{\text{model}}$ for both models: $C_{\text{model}} = \alpha_1 T_{\text{model}} + \alpha_2 M_{\text{model}} + \alpha_3 S_{\text{model}}$

  Compute total evaluation cost $V_{\text{model}}$ for each model: $V_{\text{model}} = J_{\text{model, total}} + \lambda C_{\text{model}}$
  Update trust-weight $\beta_{t+1} = \beta_t + \eta_t(V_{\text{i-ResNet}} - V_{\text{DipDNN}})$
**end for**

---

In the following, we summarize the setups of parameters in implementation.

- $J_{\text{threshold}}$ indicates the acceptable value range of the inverse learning performance in certain tasks. The value is determined by cross-validation to prevent being trapped in the local optimum when training invertible models. It is used as we observe both models may experience trivial convergences in training.

- Weighting factors $\lambda_i, I = 1, 2, 3$ are the most important to the switching results and are task-specific to combine the three metrics of inverse learning. Without prior knowledge of the datasets, we use equal weights ($\lambda_1 = \lambda_2 = \lambda_3$).

- We set the number of iterations for switching $T = 500$ to allow the model to fully adapt.

- The weights $\alpha_1, \alpha_2$ balance different components in the combination of computational cost: time complexity (forward and inverse computation time) and model size (parameter memory that mainly depends on NN depth in inverse problems).

- $\eta_t = 0.01$ $\eta_t$ is used to control the speed of updating the trust-weight parameter $\beta$ by evaluating i-ResNet and DipDNN. In our implementation, $0.01$ is empirically found stable in adaptation.

Specifically, we adopt trust-weight $\beta$ to make the switching process stable and converge to an optimal selection, which has been proven to improve online control in switching agents. $\beta$ is updated based on the difference in the combined evaluations of i-ResNet and DipDNN.

For image classification tasks, the algorithm gravitates towards i-ResNet because of its lower computational cost and acceptable accuracy, reflected by the trust-weight parameter favoring i-ResNet. For tasks requiring exact inversion, such as lossless data hiding or power system state estimation, the

algorithm shifts towards DipDNN due to its ability to provide a more accurate inverse, which results in a consistently lower evaluate function value for DipDNN.

## A.7 NUMERICAL RESULTS

Table 1: Comparison of ResNet, I-ResNet, and DipDNN for MNIST Classification

| Model | Config. | Acc. |
|---|---|---|
| ResNet | Conv | 97.89% |
| | MLP | 97.65% |
| I-ResNet | Conv | 96.32% |
| | MLP | 97.20% |
| DipDNN | MLP | 97.69% |

Table 2: Comparison of ResNet, I-ResNet, and DipDNN for CIFAR-10 Classification

| Model | Block | Channels | Accuracy (CIFAR-10) |
|---|---|---|---|
| ResNet - Conv | 12 | 16, 64, 256 | 89.64% |
| | 12 | 64, 64, 64 | 89.21% |
| | 12 | 16, 64, 256 | 90.79% |
| | 39 | 12, 12, 12 | 88.06% |
| | 39 | 12, 48, 192 | 90.69% |
| I-ResNet - Conv | 12 | 64, 64, 64 | 83.22% |
| | 12 | 16, 64, 256 | 87.79% |
| | 39 | 12, 12, 12 | 84.69% |
| | 39 | 16, 64, 256 | 90.00% |
| | 39 | 12, 48, 192 | 90.26% |
| DipDNN - Conv | 12 | 12, 12, 12 | 84.29% |
| | 39 | 12, 12, 12 | 88.06% |
| | 39 | 12, 48, 192 | 90.69% |
| ResNet - MLP | 12 | - | 66.97% |
| I-ResNet - MLP | 12 | - | 66.69% |

Table 3: Comparison of ResNet, i-ResNet, and DipDNN for physical system tasks.

(a) ResNet vs. DipDNN

| | ResNet | ResNet $k = 1$ | ResNet $k = 2$ | DipDNN |
|---|---|---|---|---|
| **NS** | 0.00022 | 0.01498 | 0.0553 | 0.1258 |
| **Power Flow** | 5.79e-07 | 1.84e-05 | 3.19e-05 | 2.01e-06 |

(b) i-ResNet vs. DipDNN

| | i-ResNet | i-ResNet $k = 1$ | i-ResNet $k = 2$ | DipDNN |
|---|---|---|---|---|
| **NS** | 0.1635 | 7.7005 | 0.3343 | 0.1258 |
| **Power Flow** | 5.73e-05 | 2.08e-05 | 6.95e-05 | 2.01e-06 |

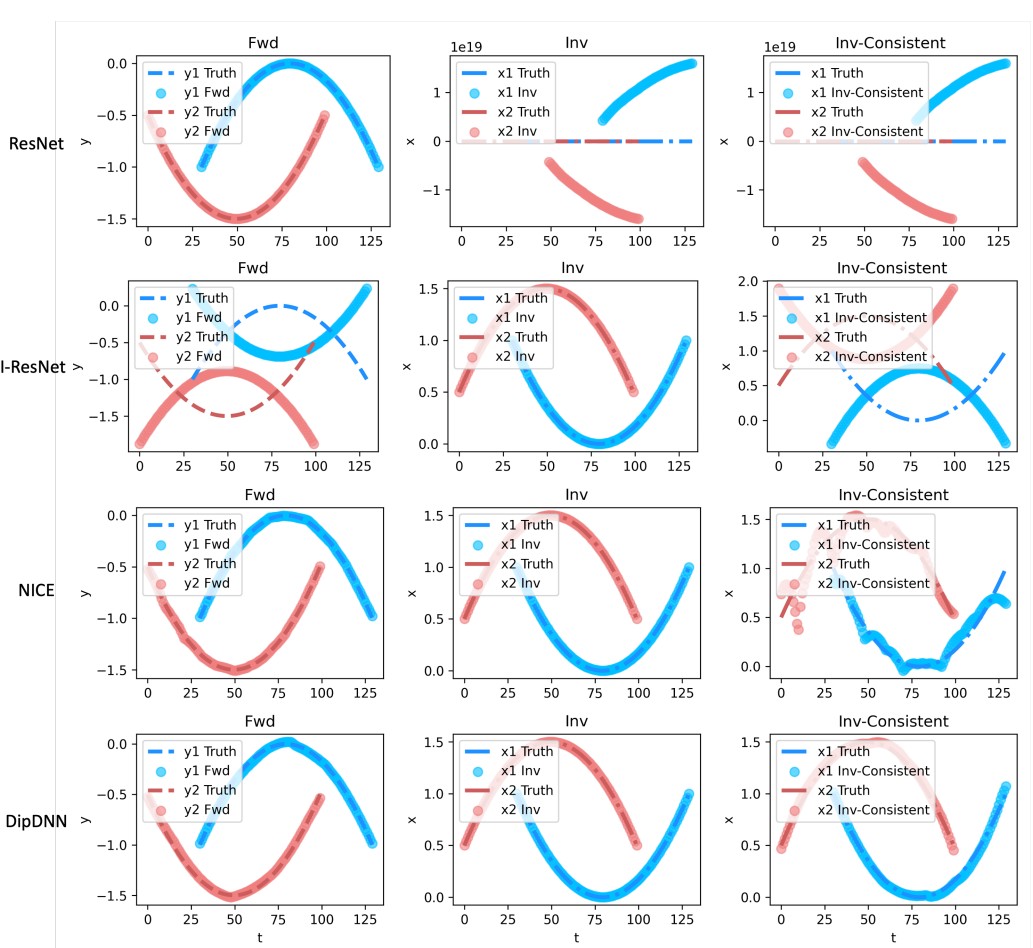

Figure 10: Fwd and Inv approximation on synthetic data.

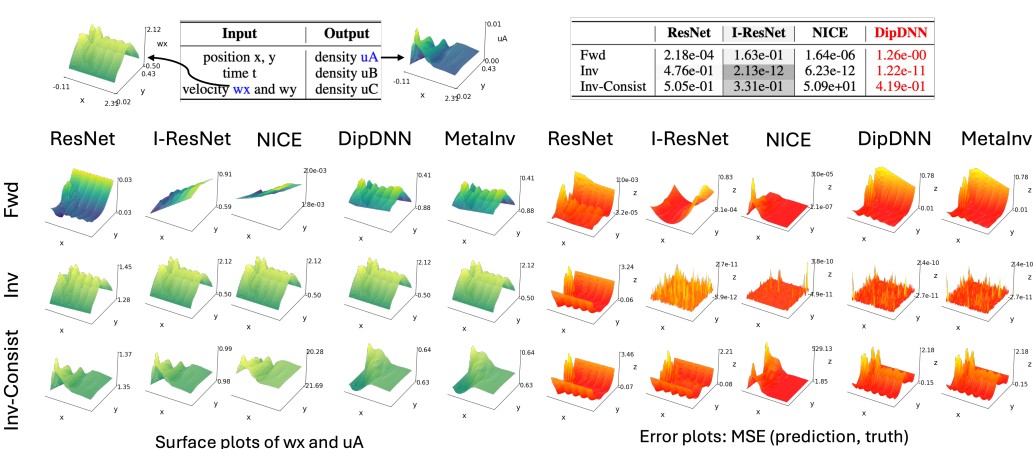

Figure 11: Forward and inverse performance of different models on Navier-Stokes flow dynamics.

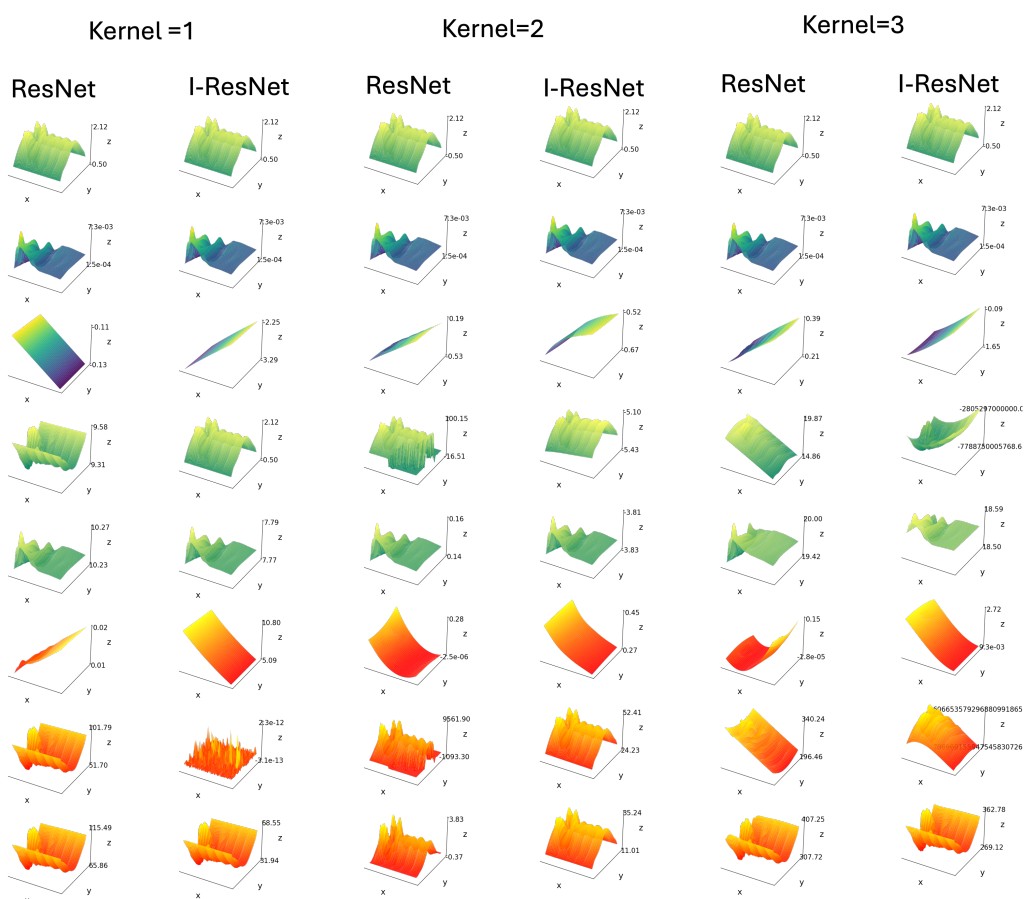

Figure 12: Performance of convolutional models on Navier-Stokes flow dynamics.

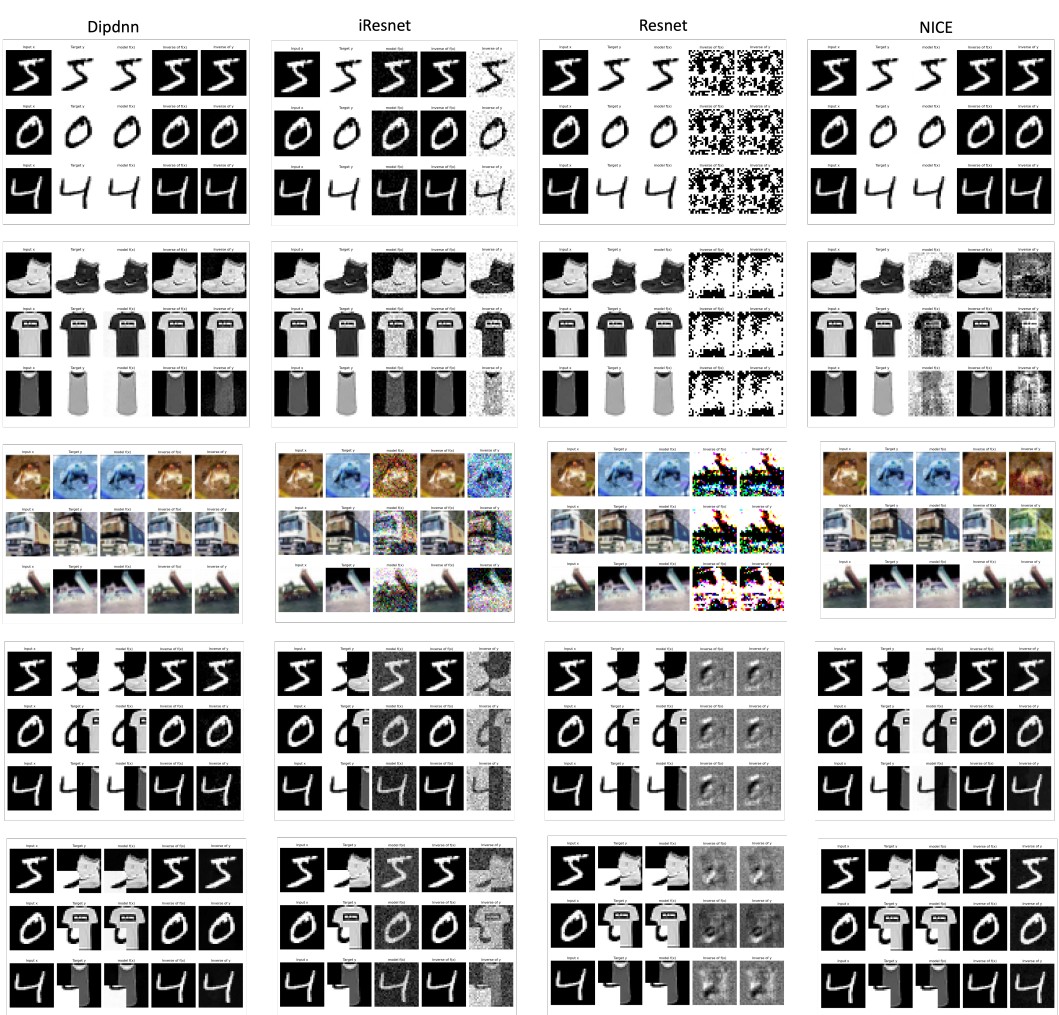

Figure 13: Performance of different models on different image convert tasks.

