# OpenReview forum: "MetaInv: Overcoming Iterative and Direct Method Limitations for Inverse Learning"
_ICLR.cc/2025/Conference — Submitted to ICLR 2025_

### Official Review · Reviewer_aJKp · 2024-10-31

**Soundness:** 3
**Presentation:** 3
**Contribution:** 2
**Rating:** 3
**Confidence:** 4

**Summary:**

This paper analysed the limitations of two specific invertible models, i-ResNet and DipDNN, and proposed a selection mechanism to choose one of the two models for a task. The basis of selection is essentially selecting the best performing model, preferring one model if neither is good.

**Strengths:**

- The authors did a good review of two prior models and proposed an interesting comparison of the two models.
- The writing is easy to follow.

**Weaknesses:**

- The work lacks strong motivation. Why two specific models were chosen to be analysed? Why do we need a single meta-model to work on tasks of a completely different nature? To me, it is quite obvious that we need different models for image data and physics data. The authors should provide a convincing argument to justify their work.
- It is not clear why only i-ResNet and DipDNN are chosen to be analysed. It would be better to aim for generalisable analyses that provide insight into a class of solutions.
- The main proposed algorithm (MetaInv) is not clearly described in the main text. More importantly, there is insufficient justification for the preference rules.

**Questions:**

- "MetaInv" is in the title, but the algorithm is not in the main text. Is there a reason to put it in the appendix?
- In 3.1, Inv error, should x_i be y_i instead?
- Figure 8 is difficult to understand. Maybe provide a better description in the caption?

---

> ### Author Response · Authors · 2024-11-25
>
> We appreciate the reviewer for pointing out the writing issues, and provide responses to the questions below.
>
> **Question on MetaInv:**
> The algorithm was included in the Appendix due to the space limit. We have moved it back to the main part and reduced the less important contents.
>
> **Question on Sec. 3.1:**
> We have not only made the corrections but also rewritten Sec. 3.1 for clarification and better readability. The revision includes:
>
> Inverse problems span numerous applications that involve recovering original variables from observed outputs. This work focuses on inverse learning through invertible mapping recovery, which is for point estimates of images or physical system states. The problem is typically formulated as approximating a forward mapping $f_{\theta}: \mathbb{R}^n \to \mathbb{R}^n$, where $y = f_{\theta}(x)$ is invertible. The goal is to find the relative inverse mapping $g_{\vartheta}$ such that $x = g_{\vartheta}(y) = f_{\theta}^{-1}(y)$, ensuring consistency with the forward process. The demands for two-way mapping rule recovery are distinct and varied even in one task. Thus, this work evaluates them using the following performance metrics:
>
> - **Forward Prediction Error (Fwd):** Same as common one-way learning, it measures the ability of the model to predict $y$ from $x$. The invertible model is trained by minimizing the forward prediction loss (any discriminative learning loss $\ell_{\text{fwd}}$):  $f^* = \arg \min_{\theta}\ell_{\text{fwd}}$  $({y},f(x))$
>
> - **Inverse Reconstruction Error (Inv):** The reconstruction error evaluates the model's invertibility to recover the original inputs (where $\ell_{\text{inv}}$ is the mean square error for point estimates):  $\ell_{\text{inv}} (x, g_{\vartheta} (f (x)))$.
>
> - **Inverse Consistency (Inv-Consist):** It assesses the consistency between forward and inverse mappings by comparing the inverse predictions with the true labels ${y}$, rather than just the forward outputs:  $\ell_{\text{fwd}} (x, g_{\theta} (y))$.
>
> It is important to note the distinction between **inverse accuracy** and **consistency**. Inverse accuracy, or reconstruction error, indicates only the model's invertibility to compute the inverse, where analytical invertibility yields an error-free result, and numerical invertibility minimizes round-off errors. Consistency, on the other hand, evaluates the model's capability of two-way learning. It reveals the precision of global inversion. For example, minimizing reconstruction error doesn't necessarily need a low forward approximation error, while the consistency error integrates errors from both forward and inverse processes. Most works involving approximate one-to-one mappings focus on inverse accuracy alone, such as in image classification and recovery.
>
> **Question on Fig. 8:**
>
> We provided a detailed explanation of Fig. 8 (now Fig. 7 in the revised manuscript) below and modified the caption in the paper correspondingly.
>
> Fig. 7 presents the performance comparison of different methods (ResNet, i-ResNet, NICE, DipDNN, and MetaInv) on the power flow case. The dataset is derived from an 8-bus power system, where load inputs ($x$) are used to estimate voltages ($y$) at different buses for forward mapping, and the inverse is the recovery of load conditions given voltages. Accurate inverse learning is essential here for recovering underlying system physics and ensuring consistency in power flow analysis.
>
> The top-left plot shows forward voltage predictions (Fwd), and the two bottom plots show the inverse load recovery for both reconstruction and consistency (Inv and Inv-Consist). The table (top-right) provides quantitative error metrics (Fwd, Inv, and Inv-Consist) for the evaluated methods. While i-ResNet shows tightly bounded errors due to its Lipschitz constraint ($\text{Lip}<1$), its voltage predictions (top-left) reveal a pattern that diverges from the ground truth. This is attributed to the contractive property, which limits pointwise accuracy.
>
> In contrast, DipDNN achieves better fits for both forward and inverse tasks. DipDNN, selected by MetaInv in this case, captures the system's essential behaviors more effectively, as evident from the closer alignment of its predictions with the ground truth. The inverse consistency plot (bottom-right) highlights DipDNN and MetaInv’s superior performance in achieving forward-inverse alignment compared to i-ResNet. However, excessive contraction, such as from convolutional layers, can distort inverse mappings, especially in systems with low redundancy.

---

> ### Author Response · Authors · 2024-11-25
>
> We appreciate the reviewer’s concerns regarding the motivation of our work and the selection of specific models for analysis. Based on previous responses, we clarify how this study narrows its scope, justifies the choice of models, and provides theoretical and empirical contributions to inverse problems.
>
> This study focuses on a specific class of inverse problems requiring **deterministic two-way mappings** for tasks such as physical system state estimation, image classification and recovery, and adaptive control. Unlike probabilistic setups or generative learning methods, which aim to model multimodal distributions or perform density estimation, we aim to recover **point estimates with forward-inverse consistency**. This focus prioritizes precision and interpretability, making explicit invertibility a central requirement for the model design.
>
> We agree with the reviewer that there is a general difference between image and physics data. However, other than the data type, more factors and scenarios need to be considered when solving inverse problems:
>
> - **Data types and inverse learning tasks:** Even using the same image data (e.g., MNIST and CIFAR10), there can be significantly different tasks. A major group of existing works on inverse problems focuses on density estimation tasks, which aim to generate complex distributions of images and fall under generative learning in a probabilistic setup. However, there are also image classification and recovery tasks, which belong to discriminative learning and require precise point estimates. Similarly, physical systems involve both stochastic modeling to quantify uncertainties and deterministic inverse/two-way mappings to estimate precise system states.
>
> - **Application-specific factors:** Beyond data type, specific characteristics of application cases play a critical role in model performance. These include but are not limited to: i) the complexity of the underlying mapping, ii) the information redundancy of inputs, and iii) the precision requirements for forward and inverse predictions. Theorems 1 and 2 analyze the limitations of i-ResNet for i). Theorem 3 analyzes the transformation equivalency for ii), as the convolution layer compresses information. These are validated by Fig. 3, Fig. 9, Fig. 12, and Table II using different kernels of convolutions (different information compressions). The three metrics in Sec. 3.1 are defined to quantify for iii).  By analyzing these diverse aspects comprehensively, our study evaluates the trade-offs of i-ResNet and DipDNN both theoretically and empirically.

---

> > ### Author Response · Authors · 2024-11-25
> >
> > Based on our reviews, especially of papers citing previous SOTA methods such as NICE, RealNVP, and i-ResNet, the literature includes:
> >
> > 1. **Invertible models applicable to probabilistic setups and generative tasks** [2,3,5].
> > 2. **Works applying SOTA methods to different applications with minor, application-specific changes** [1,6,7,9].
> >    - For example, [6] applies NICE to image steganography, [7] applies NICE to Markov Chain Monte Carlo, and [9] uses a variation of NICE/RealNVP under stochastic modeling for physical system tasks.
> > 3. **Works solving inverse problems that do not exhibit invertibility or deterministic two-way mappings** [4,8].
> >
> > A rigorous analysis of foundational invertible architectures is central to our work. Thus, rather than providing exhaustive experimental comparisons across all existing methods, the two key architectures from existing works are
> >
> > - **Affine coupling layers** (used in NICE and its extended versions): These layers provide analytical invertibility and computational efficiency but are limited by their reliance on affine transformations and split variables. DipDNN addresses these limitations by relaxing constraints while preserving analytical invertibility.
> >
> > - **Residual layers with Lipschitz constraint** (used in i-ResNet): These models are built on residual networks to enhance forward approximation capacity. Unlike affine-coupling layers, they enforce invertibility via Lipschitz continuity, with the inverse computed iteratively through a fixed-point algorithm.
> >
> > These architectures were chosen because they represent distinct ways to achieve invertibility: one through analytical construction and the other through constraints on Lipschitz continuity. By rigorously analyzing these models, we provide insights into their strengths and weaknesses across various deterministic inverse problems.
> >
> >
> > **References**
> > [1] Ardizzone, Lynton, et al. "Guided image generation with conditional invertible neural networks." arXiv preprint arXiv:1907.02392 (2019).
> >
> > [2] Bishop, Christopher M. "Mixture density networks." (1994).
> >
> > [3] Han, Xintian, Mark Goldstein, and Rajesh Ranganath. "Survival mixture density networks." Machine Learning for Healthcare Conference. PMLR, 2022.
> >
> > [4] Xu, Peng, et al. "Inverse design of a metasurface based on a deep tandem neural network." JOSA B 41.2 (2024): A1-A5.
> >
> > [5] Kingma, Durk P., et al. "Improved variational inference with inverse autoregressive flow." Advances in neural information processing systems 29 (2016).
> >
> > [6] Zhang, Zhuo, Hongjun Wang, and Jia Liu. "A Method for Image Steganography based on NICE Model." 2022 International Conference on Machine Learning, Cloud Computing and Intelligent Mining (MLCCIM). IEEE, 2022.
> >
> > [7] Song, Jiaming, Shengjia Zhao, and Stefano Ermon. "A-nice-mc: Adversarial training for mcmc." Advances in neural information processing systems 30 (2017).
> >
> > [8] Habring, Andreas, and Martin Holler. "Neural-network-based regularization methods for inverse problems in imaging." GAMM-Mitteilungen (2024): e202470004.
> >
> > [9] Ardizzone et al. "Analyzing inverse problems with invertible neural networks." ICLR 2019

---

### Official Review · Reviewer_yNgD · 2024-11-03

**Soundness:** 1
**Presentation:** 1
**Contribution:** 1
**Rating:** 3
**Confidence:** 4

**Summary:**

The authors propose a method, termed MetaInv, to dynamically choose between two approaches for inverse modeling depending upon which one is most appropriate for the problem.

**Strengths:**

1) Inverse problems are important, and therefore the general topic is significant

**Weaknesses:**

1) The presentation is often poor.  The introduction is often vague/unclear regarding existing methods and their weaknesses.  There are major mathematical typos:  e.g., the equation in Line 147 does not seem to be valid.  Although I can guess, the authors never describe what l_{2} is, and what exactly are we summing over in the equations from lines 139-155.

2) The review of the literature on inverse modeling misses large portions of the literature. Some examples include (but may not be limited to) genetic algorithms, conditional Invertible neural networks, mixture density networks, and the so-called Tandem model.   The authors ought to mention some of these approaches and explain why they were not applicable for this study, or at least why they were not included in experiments.   The authors can find examples of these existing methods/publications in Ardizzone et al. "Analyzing inverse problems with invertible neural networks." arXiv preprint arXiv:1808.04730 (2018),    or publications that cite this one.

3) The authors show that their method is beneficial for just two methods in the literature: DipDNN and iResNet.  This becomes interesting if the result of combining these two approaches with MetaInv yields a method that achieves state-of-the-art capabilities (e.g., inverse accuracy, or computational efficiency).  However, the authors only show that their MetaInv approach leads to better/comparable performance to DipDNN and iResNet.  Therefore it is unclear whether the result of all this work improves state-of-the-art in any way or just improves these two methods.  Although improvement over these two particular methods is still interesting as a proof-of-concept for the MetaInv idea, in my opinion it is not sufficiently significant/interesting for this venue.

**Questions:**

I welcome a response to comment (2) in the "weaknesses" section, although I request that the authors only respond if their response is clearly written, and addresses in a clear/compelling way (i) why so much existing work on inverse modeling was omitted from consideration, and (ii) why this work is still significant despite the exclusion of so much existin work.

---

> ### Author Response · Authors · 2024-11-25
>
> **Comment:** Mathematical typos from lines 139-155.
>
> Thanks for pointing out mathematical typos. We have not only made the corrections but also rewritten Sec. 3.1 for clarification and better readability. The revision includes:
>
> Inverse problems span numerous applications that involve recovering original variables from observed outputs. This work focuses on inverse learning through invertible mapping recovery, which is for point estimates of images or physical system states. The problem is typically formulated as approximating a forward mapping $f_{\theta}: \mathbb{R}^n \to \mathbb{R}^n$, where $y = f_{\theta}(x)$ is invertible. The goal is to find the relative inverse mapping $g_{\vartheta}$ such that $x = g_{\vartheta}(y) = f_{\theta}^{-1}(y)$, ensuring consistency with the forward process. The demands for two-way mapping rule recovery are distinct and varied even in one task. Thus, this work evaluates them using the following performance metrics:
>
> - **Forward Prediction Error (Fwd):** Same as common one-way learning, it measures the ability of the model to predict $y$ from $x$. The invertible model is trained by minimizing the forward prediction loss (any discriminative learning loss $\ell_{\text{fwd}}$):  $f^* = \arg \min_{\theta}\ell_{\text{fwd}}$  $({y},f(x))$
>
> - **Inverse Reconstruction Error (Inv):** The reconstruction error evaluates the model's invertibility to recover the original inputs (where $\ell_{\text{inv}}$ is the mean square error for point estimates):  $\ell_{\text{inv}} (x, g_{\vartheta} (f (x)))$.
>
> - **Inverse Consistency (Inv-Consist):** It assesses the consistency between forward and inverse mappings by comparing the inverse predictions with the true labels ${y}$, rather than just the forward outputs:  $\ell_{\text{fwd}} (x, g_{\theta} (y))$.
>
> It is important to note the distinction between **inverse accuracy** and **consistency**. Inverse accuracy, or reconstruction error, indicates only the model's invertibility to compute the inverse, where analytical invertibility yields an error-free result, and numerical invertibility minimizes round-off errors. Consistency, on the other hand, evaluates the model's capability of two-way learning. It reveals the precision of global inversion. For example, minimizing reconstruction error doesn't necessarily need a low forward approximation error, while the consistency error integrates errors from both forward and inverse processes. Most works involving approximate one-to-one mappings focus on inverse accuracy alone, such as in image classification and recovery.

---

> ### Author Response · Authors · 2024-11-25
>
> **Comment:** Why several inverse learning methods were omitted?
>
> The literature on inverse modeling spans a broad spectrum, covering diverse applications and inverse models, as the reviewer highlights with the additional references [1]-[4]. Our study focuses on **deterministic two-way/inverse mapping** for inverse problems. Specifically, we target the invertibility design of models for tasks requiring point estimates, such as physical system state estimation, image classification and recovery, and adaptive control. These inverse problems differ from **generative tasks** like image density estimation, which form many previous works in the computer science community. Consequently, we evaluate invertible models' performances in **discriminative learning**, emphasizing forward approximation, inverse reconstruction, and forward-inverse consistency.
>
> The methods mentioned by the reviewer either rely on stochastic modeling for image generation or lack explicit invertibility. Below, we discuss these methods in detail and justify their limited relevance to our work:
>
> **Genetic Algorithms**
> Genetic algorithms are widely used in optimization and design problems but are computationally intensive and often impractical for high-dimensional nonlinear inverse tasks. Additionally, they do not inherently provide invertibility or forward-inverse consistency, making them unsuitable for the deterministic two-way mappings we aim to achieve.
>
> **Conditional Invertible Neural Networks (cINNs) [1]**
> Conditional Invertible Neural Networks (cINNs) extend NICE and RealNVP models by enforcing invertibility through affine coupling layers. These models are primarily designed for generative tasks, using maximum likelihood principles to improve image density estimation. While cINNs perform well in generative tasks, they are less suited for tasks involving point estimates. Furthermore, we have reviewed, analyzed, and tested the NICE model. Since cINNs' analytical invertibility relies on affine coupling layers, similar to NICE, the comparison of the targeted inverse problem has been covered in this work.
>
> **Mixture Density Networks (MDNs) [2,3]**
> Mixture Density Networks (MDNs) are designed for probabilistic inverse modeling, where the outputs are characterized by multimodal distributions. These models preserve tractable probability density and involve uncertainty quantification, but they are not applicable to recover deterministic mappings that ensure one-to-one consistency between input and output. Furthermore, MDN models do not inherently enforce invertibility, which is central to our approach.
>
> **Tandem Models [4]**
> The Tandem models are specifically designed to solve inverse design problems, e.g., for optical materials. They focus on two-stage optimization, coupling an inverse-design network with a pre-trained forward model, using the latter to supervise the training of the former. Usually, the forward model is known, and the two models have different architectures. Although the general scheme involves inverse modeling, no invertibility is enforced in the Tandem structure. However, since Tandem models also target point estimates, such an inverse design problem could be an application case of our targeted inverse problem.

---

> > ### Author Response · Authors · 2024-11-25
> >
> > Based on our reviews, especially of papers citing previous SOTA methods such as NICE, RealNVP, and i-ResNet, the literature includes:
> >
> > 1. **Invertible models applicable to probabilistic setups and generative tasks** [2,3,5].
> > 2. **Works applying SOTA methods to different applications with minor, application-specific changes** [1,6,7].
> >    - For example, [6] applies NICE to image steganography, and [7] applies NICE to Markov Chain Monte Carlo.
> > 3. **Works solving inverse problems that do not exhibit invertibility or deterministic two-way mappings** [4,8].
> >
> > For example, the work by Ardizzone et al. ("Analyzing inverse problems with invertible neural networks." ICLR 2019) mentioned by the reviewer uses a variation of NICE/RealNVP under stochastic modeling to resolve physical system estimation tasks.
> >
> > **References**
> > [1] Ardizzone, Lynton, et al. "Guided image generation with conditional invertible neural networks." arXiv preprint arXiv:1907.02392 (2019).
> >
> > [2] Bishop, Christopher M. "Mixture density networks." (1994).
> >
> > [3] Han, Xintian, Mark Goldstein, and Rajesh Ranganath. "Survival mixture density networks." Machine Learning for Healthcare Conference. PMLR, 2022.
> >
> > [4] Xu, Peng, et al. "Inverse design of a metasurface based on a deep tandem neural network." JOSA B 41.2 (2024): A1-A5.
> >
> > [5] Kingma, Durk P., et al. "Improved variational inference with inverse autoregressive flow." Advances in neural information processing systems 29 (2016).
> >
> > [6] Zhang, Zhuo, Hongjun Wang, and Jia Liu. "A Method for Image Steganography based on NICE Model." 2022 International Conference on Machine Learning, Cloud Computing and Intelligent Mining (MLCCIM). IEEE, 2022.
> >
> > [7] Song, Jiaming, Shengjia Zhao, and Stefano Ermon. "A-nice-mc: Adversarial training for mcmc." Advances in neural information processing systems 30 (2017).
> >
> > [8] Habring, Andreas, and Martin Holler. "Neural-network-based regularization methods for inverse problems in imaging." GAMM-Mitteilungen (2024): e202470004.

---

> ### Author Response · Authors · 2024-11-25
>
> **Comment:** Why this work remains significant?
>
> We appreciate the reviewer’s concerns regarding the weakness of existing methods addressed in our study. Based on the previous response, we clarify how this work narrows the scope and provides theoretical analysis for a unique contribution to inverse problems.
>
> This study focuses on a specific class of inverse problems requiring deterministic two-way mappings for tasks like physical system state estimation, image classification and recovery, and adaptive control. Unlike probabilistic setups or generative learning (e.g., MDNs, cINNs), where the objective often involves multimodal distribution density estimation, the goal here is to recover point estimates with forward-inverse consistency. Such problems prioritize precision and interpretability, necessitating explicit invertibility in the model design.
>
> Thus, existing works that exhibit invertible model architectures are central to our work, where two key architectures are:
>
> - **Affine coupling layers** (used in NICE and its extended versions): These layers provide analytical invertibility and computational efficiency, but the approximation capability is constrained by their reliance on affine transformations and split variables. DipDNN relaxes the limitations while maintaining the analytical inverse.
>
> - **Residual layers with Lipschitz constraint** (used in i-ResNet): These are built upon the residual network for forward approximation capacity. Unlike analytical inverses, they ensure invertibility by enforcing Lipschitz continuity. The inverse requires iterative computation using a fixed-point algorithm.
>
> By focusing on DipDNN and i-ResNet, we provide a rigorous analysis of foundational invertible architectures, rather than providing exhaustive experimental comparisons across all existing methods.
>
> For both theoretical analysis and empirical implementation, we identify the trade-off of the two invertible models for various deterministic inverse problems. This is why MetaInv is introduced to supplement the capabilities of the two representative approaches for arbitrary tasks. By building on and improving DipDNN and i-ResNet, MetaInv provides a unified approach that achieves consistent forward-inverse performance while maintaining computational efficiency.

---

### Official Review · Reviewer_jQ3F · 2024-11-06

**Soundness:** 3
**Presentation:** 3
**Contribution:** 3
**Rating:** 5
**Confidence:** 2

**Summary:**

This paper first provides a detailed analysis of the limitations in current invertible architectures, examining the trade-offs between iterative and analytical approaches. It then proceeds with proposing a meta-inverse framework that dynamically combines the advantages of both i-ResNet and DipDNN, by dynamically switching between them based on ask-specific signals.

**Strengths:**

A very detailed analysis of the limitations in current invertible architectures,

**Weaknesses:**

A rather short and not very extensive presentation of the switching algorithm and its associated challenges.

**Questions:**

What were the key challenges in developing the switching algorithm, and what were the main contributions here?

---

> ### Comment · Reviewer_jQ3F · 2024-11-25
> **Thank you for the answers**
>
> I will keep my score

---

> ### Author Response · Authors · 2024-11-27
> **Response to Reviewer jQ3F on the Switching Algorithm**
>
> We appreciate the reviewer’s time and efforts, and we apologize for the delay in responding to your questions about the short presentation of MetaInv. Given the importance of the switching algorithm, which is included in the title as an essential contribution, we wanted to take the time to carefully address your concerns and revise the manuscript accordingly. Below, we outline the key motivations for this work towards the hybrid method, followed by an analysis of the specific difficulties addressed during the algorithm development.
>
> This study focuses on a specific class of inverse problems requiring deterministic two-way mappings for tasks such as physical system state estimation, image classification and recovery, and adaptive control. Since the objective is to recover point estimates with forward-inverse consistency, it necessitates models with explicit invertibility.
>
> As the reviewer recognized, we have targeted two representative invertible architectures and presented the analysis both theoretically and empirically. Then, we found that, even with a narrowed scope, many factors and scenarios need to be considered when solving specific inverse problems. The differences can be evident when considering:
>
> - **Data types and inverse learning tasks:** Even using the same data, there can be significantly different tasks. For example, image datasets (e.g., MNIST and CIFAR10), besides the major group of density estimation, also have image classification and recovery tasks. The latter emphasizes forward classification accuracy and inverse reconstruction. Similarly, physical systems involve both stochastic modeling to quantify uncertainties and deterministic inverse/two-way mappings to estimate precise system states.
>
> - **Application-specific factors:** Beyond data type, specific characteristics of application cases play a critical role in model performance. These include but are not limited to:
>
>    - i) The complexity of the underlying mapping - nonlinearity, dimension, coupling of variables, chaotic correlation, etc.  Theorems 1 and 2 analyze the limitations of i-ResNet.
>
>    - ii) The information redundancy of inputs - if the data have sparsity to be compressed and if the information can be retained to trace back to inputs. For example, images usually have redundancy, while variables with physical meanings are independent with minimum redundancy. We have Theorem 3 and test results (Fig. 3, Fig. 9, Fig. 12, and Table II) using different convolution kernels.
>
>   - iii) The precision requirements for forward and inverse predictions, which are quantified by three metrics in Sec. 3.1.
>
>
> These application-specific factors, combined with the trade-offs of invertible architectures, motivate the idea of switching algorithms. We encountered difficulties in balancing the objectives of inverse learning and dynamically adapting to task-specific characteristics. Therefore, we designed the **trust-weighted switching mechanism**, a hybrid approach inspired by learning-augmented online control.
>
> From (i)–(iii), the objectives of inverse learning, such as high expressive power, strict invertibility, low redundancy, bi-directional consistency, and computational efficiency, are often in conflict. For example, i-ResNet’s flexibility enables it to model complex mappings but may sacrifice precision in inverse mappings.  DipDNN’s analytical invertibility minimizes inverse error but limits flexibility, particularly for tasks involving data redundancy.
>
> To address these challenges, the switching mechanism incorporates the trust-weight $\beta$, which integrates *forward accuracy*, *inverse accuracy*, *inverse consistency*, and *computational cost* into a single evaluation function:
> $
> V_{\text{model}} = J_{\text{model, total}} + \lambda C_{\text{model}}.
> $
>
> The evaluation of these factors is not fixed during learning. To ensure smooth switching and avoid trivial convergences, \(\beta\) is updated dynamically:
> $
> \beta_{t+1} = \beta_t + \eta_t (V_{\text{i-ResNet}} - V_{\text{DipDNN}}),
> $
> where $\eta_t$ is the learning rate (empirically set to 0.01 for stability); additionally, a parameter $J_{\text{threshold}}$ sets an acceptable performance range, ensuring robustness to data uncertainties such as noise. This parameter is crucial, as inverse learning is highly sensitive to such uncertainties.
>
>
> The following revisions are made:
>
> The MetaInv algorithm was included in the Appendix due to the space limit. We have moved it back to the main part and reduced the less important contents.
>
> We revised the corresponding description of MetaInv in Sec. 4.2, with a special focus on the difficulties and contributions of the switching algorithm, as highlighted by the reviewer.
>
> The details of implementing the algorithm are included in the Appendix, e.g., parameter setups.

---

### Meta-Review · Area_Chair_aHDQ · 2024-12-17

**Metareview:**

The paper studies invertible neural network models in the context of invertible prediction tasks with image, power system measurement and fluid dynamics data. The authors provide a detailed analysis of tradeoffs, advantages and limitations of two models: i-ResNet [1] and DipDNN [2] representing iteratively and analytically invertible models respectively. In particular, the authors identify three metrics to consider: forward prediction error, inverse reconstruction error and inverse consistency. Then, the authors propose MetaInv, a method for adaptively deciding on which of the two architectures to choose for a given task. The authors show empirically (1) the expressivity limitations of i-ResNet, (2) trade-offs in performance between different invertible methods, (3) MetaInv can correctly choose which invertible model to use.

Strengths:
- Interesting discussion on the tradeoffs between different invertible architectures
- Interesting insights into the limitations of i-ResNet
- MetaInv method that can decide between competing invertible models in a practical setting
- Experiments on diverse domains

Weaknesses:
- The choice of specific models (i-ResNet and DipDNN) seems arbitrary
- Proposed method is practical, but not particularly interesting from a scientific point of view, as it is deciding on which method to use based on a combination of performance metrics
- Reviewers highlighted that the presentation is poor
- The results do not show an improvement to state-of-the-art on an established benchmark

Decision recommendation:
While the paper provides interesting insights, the methodological novelty is limited and improvements need to be made to the presentation. I would recommend that the authors should improve the presentation and better explain the motivation for the work and the design decisions made in the paper. It would also help to include results on established benchmarks with baselines evaluated in prior work. The reviewers unanimously rejected the paper. I thus recommend a rejection.

[1] Jens Behrmann, Will Grathwohl, Ricky TQ Chen, David Duvenaud, and Jorn-Henrik Jacobsen. ¨
Invertible residual networks.

[2] Jingyi Yuan, Yang Weng, and Erik Blasch. Dipdnn: Preserving inverse consistency and approximation
efficiency for invertible learning

**Additional Comments On Reviewer Discussion:**

The reviewers unanimously rejected the paper highlighting poor presentation, unclear motivation for some of the design decisions, and lack of improvement to state-of-the-art on established benchmarks. During the rebuttal phase, the authors updated the presentation, and in particular added an explanation of the MetaInv method to the main text and expanded on the motivation for the method. However, reviewers were not satisfied with these updates, and did not increase their scores.

---

### Decision · Program_Chairs · 2025-01-22

Reject